# OpenReview forum: "Avalon's Game of Thoughts: Battle Against Deception through Recursive Contemplation"
_ICLR.cc/2024/Conference — ICLR 2024 Conference Withdrawn Submission_

### Official Review · Reviewer_p3aL · 2023-10-24

**Soundness:** 3 good
**Presentation:** 3 good
**Contribution:** 3 good
**Rating:** 6
**Confidence:** 3

**Summary:**

The paper explores the challenges LLMs face in processing deceptive information, utilizing the game Avalon for evaluation. It demonstrates that incorporating human-like cognitive patterns improves LLMs' performance in the game. The authors introduce "ReCon", a method that merges "formulation and refinement contemplation" with "first- and second-order perspective transitions" for LLM prompts. In practice, this involves a phase in which the LLM is asked to contemplate internally from a first-order (its own) perspective, and then to contemplate again from a second-order (other players') perspective. This method's efficacy is validated using ChatGPT and Claude in the Avalon game.

**Strengths:**

The results are interesting, the paper is well written, and the analysis is reasonable. The work is also novel, I believe, and it improves our understanding about how to make LLMs safer.

**Weaknesses:**

My main comment is that I would like to see how the proposed approach may be applied in case of attempts to extort PII from LLMs or jailbreaking outside of the Avalon environment. It should be fairly straightforward to assess this. Have the authors tried it or can they comment on it at least?

Other comments:

* The paper argues that ReCon is superior to CoT and its variants that lack either contemplation or perspective transition. I think that the results are pretty convincing, although I want to see some confidence intervals for the estimates in Figure 4. One inconsistency I noted, which the authors also mention, is the differing performance between ReCon without refinement contemplation and ReCon without formulation contemplation when applied to Claude versus ChatGPT. Some insight into this discrepancy would be useful. It might also be beneficial to contrast paired strategies, such as ReCon with and without refinement contemplation, to reinforce the findings. Have the authors tried this approach?

* Another question is regarding the scalability of LLMs: Would CoT match ReCon's performance when applied to larger LLMs?

* The number of games used to generate Figure 4 is unclear. It's unclear whether Figure 4's results are derived from the 20 full Avalon games mentioned in the 4.2 Setup, especially since multiple strategies were simultaneously deployed for a single side (good/evil) in this context so that's likely not the case. Clarification on this, along with the inclusion of confidence intervals for the results mentioned above, would provide readers with a clearer understanding of the robustness of the findings.

* The statement "we use ReCon for the good side" seems inconsistent and should likely read "we use CoT for the good side," correct? The paper also contains a few typographical errors that need rectification.

* It would be beneficial for the authors to disclose the specific prompts used for GPT-4 across the various evaluation metrics. Furthermore, it's worth pondering whether the results would remain consistent if humans, rather than CoT, played the game. Lastly, why omitting Claude's performance across all evaluation metrics.

**Questions:**

Mentioned above.

---

> ### Author Response · Authors · 2023-11-17
> **Rebuttal to Reviewer p3aL**
>
> Thank you for your careful review and constructive suggestions, which have helped us improve our manuscript. We are pleased to address your concern as follows.
>
> > Q1: The paper argues that ReCon is superior to CoT and its variants that lack either contemplation or perspective transition. I think that the results are pretty convincing, although I want to see some confidence intervals for the estimates in Figure 4.
>
> We are grateful for your suggestion and have addressed your concern in `Global Q2`. For more details, please see `Global Q2` in the [Global Responses](https://openreview.net/forum?id=fvhJu0FODp&noteId=Y7twEtlQGy).
>
>
>
> ---
>
>
>
> > Q2: One inconsistency I noted, which the authors also mention, is the differing performance between ReCon without refinement contemplation and ReCon without formulation contemplation when applied to Claude versus ChatGPT. Some insight into this discrepancy would be useful.
>
> Thank you for pointing out this aspect of our study. The discrepancy you noticed in Figure 4 could likely be **due to the different abilities of LLMs to hide internal thoughts from their spoken outputs**. Our empirical analysis of the game logs suggests that ChatGPT, particularly GPT-4, is more adept at concealing internal thoughts from spoken outputs than Claude.
>
> As a result, for hiding internal thoughts in spoken content, **the refinement contemplation based on GPT-4 may be more effective than the formulation process used in ReCon with ChatGPT. Conversely, the formulation contemplation is more critical for ReCon when implemented with Claude.** Thus, in Figure 4, the relative magnitude of the performance without refinement contemplation (the green bin) compared to the performance without formulation contemplation (the red bin) is different between ChatGPT and Claude.
>
>
>
> ---
>
>
>
> > Q3: It might also be beneficial to contrast paired strategies, such as ReCon with and without refinement contemplation, to reinforce the findings. Have the authors tried this approach?
>
> Thank you for your suggestion. Yes, we have carried out the ablation study you referred to, as depicted in Figure 4 (comparing the purple bin with the green bin) and in Figure 5(b) of our paper.
>
>
>
> ---
>
>
>
> > Q4: Another question is regarding the scalability of LLMs: Would CoT match ReCon's performance when applied to larger LLMs?
>
> We appreciate your interest in the scalability of large language models. We agree that this is an important and relevant question to investigate.
>
> Our opinion is that in certain situations, particularly those involving deceptive cognitive processes like in the Avalon game, **CoT would not be as effective as ReCon**. This is because an LLM agent using CoT is akin to a thoughtful person whose thoughts are fully exposed and lacks the ability to take on different perspectives. In contrast, an LLM agent with ReCon can be likened to a thoughtful person who can separate internal thoughts from spoken content and possesses the capability of perspective-taking, leading to more comprehensive thinking. Even with human intelligence as it stands, in environments characterized by deceptions and complex logic, a person thinking with ReCon is likely to have an advantage over one using CoT. Therefore, in the foreseeable future, even if LLMs reach human-level intelligence, ReCon is expected to outperform CoT in scenarios involving deceptions and complex logic.

---

> ### Author Response · Authors · 2023-11-17
> **(Continued) Rebuttal to Reviewer p3aL**
>
> > Q5: The number of games used to generate Figure 4 is unclear. It's unclear whether Figure 4's results are derived from the 20 full Avalon games mentioned in the 4.2 Setup, especially since multiple strategies were simultaneously deployed for a single side (good/evil) in this context so that's likely not the case. Clarification on this, along with the inclusion of confidence intervals for the results mentioned above, would provide readers with a clearer understanding of the robustness of the findings.
>
> We apologize for the lack of clarity in our paper regarding the number of games used for Figure 4. **To clarify, we conducted 20 games for each model shown in Figure 4, maintaining this number consistently across all experiments except for Claude, which involved around 40 games.** The rationale behind conducting a higher number of games for Claude is detailed in `Global Q2` of the [global responses](https://openreview.net/forum?id=fvhJu0FODp&noteId=Y7twEtlQGy). Additionally, regarding your suggestion about confidence intervals, we have included statistical tests in `Global Q2` of the [global responses](https://openreview.net/forum?id=fvhJu0FODp&noteId=Y7twEtlQGy). Thank you again for your constructive advice.
>
>
>
> ---
>
>
>
> > Q6: The statement "we use ReCon for the good side" seems inconsistent and should likely read "we use CoT for the good side," correct? The paper also contains a few typographical errors that need rectification.
>
> We apologize for the confusion caused by the lack of clarity in our paper. Indeed, it was our intention to use the ReCon method for the good side throughout our experiments.
>
> We have tried to test the good side using the CoT method against the evil side with ReCon. However, given the evil side's advantage (per [Avalon statistics](https://www.proavalon.com/statistics)), the good side consistently lost, leading to uniform results. Thus, **for more discriminative outcomes, we used ReCon on the good side when assessing methods for the evil side.**
>
> We will make this explanation clear in our revised paper.
>
>
>
> ---
>
>
>
> > Q7: It would be beneficial for the authors to disclose the specific prompts used for GPT-4 across the various evaluation metrics.
>
> Thank you for raising this meaningful question. For full transparency, we have provided all the complete prompts used with GPT-4 for evaluation in [this anonymous link](https://anonymous.4open.science/r/ICLR_Avalon_Rebuttal_ID_118/gpt4_evaluation_prompts.pdf) for your reference.
>
>
>
> ---
>
>
>
> > Q8: It's worth pondering whether the results would remain consistent if humans, rather than CoT, played the game.
>
> Thank you for your constructive question. Regarding the comparison with human players, we first acknowledge that current LLMs, even those utilizing the ReCon method, may not yet match the performance level of expert human players. This is a recognition of the inherent limitations of current LLM. Additionally, the challenge is further compounded by the lack of a comprehensive Avalon game dataset tailored for fine-tuning LLMs, which limits our ability to enhance their capabilities in this context. However, our paper demonstrates significant improvements when using ReCon in the Avalon game, a finding we believe is substantial compared to the CoT method.
>
> Another concern when considering conducting human experiments is that Avalon is a complex strategic game. **The gap in performance between novice and experienced players is substantial. In contrast, GPT-4's performance is more consistent, making our experimental results more transparent and reproducible.** However, you make an excellent point; conducting experiments with human players to verify the consistency of our results is crucial, and this will be an important focus of our future work.
>
>
>
> ---
>
>
>
> > Q9:  Why omitting Claude's performance across all evaluation metrics.
>
> Regarding the exclusion of Claude's performance, **our paper primarily focuses on ChatGPT as the representative model**. Consequently, apart from the initial experiment in Figure 4, we carried out additional analysis experiments specifically on ChatGPT. Moreover, **since our ReCon designs proved consistently effective for both ChatGPT and Claude in Figure 4, it's likely that the extra analysis on ChatGPT could mirror similar results for Claude.** Nonetheless, we acknowledge the value of including Claude's performance across all evaluation metrics and plan to thoroughly analyze Claude's performance in our future research.
>
> ---
>
> Thank you once again for your valuable feedback and advice. We look forward to further discussion to refine our work.

---

> > ### Author Response · Authors · 2023-11-23
> > **A Kindly Request for Response to Our Rebuttal**
> >
> > Dear Reviewer p3aL,
> >
> > With less than 6 hours remaining before the ICLR author-reviewer discussion concludes, we kindly request your prompt feedback on our rebuttal. Your insights are invaluable to us, and a timely response would be greatly appreciated to ensure a comprehensive review process.
> >
> > Additionally, we've refreshed the file providing the prompts utilized for GPT-4 evaluation, which can be accessed at this [anonymous link](https://anonymous.4open.science/r/ICLR_Avalon_Rebuttal_ID_118/gpt4_evaluation_prompts.pdf).
> >
> > If our response has addressed your concerns, we hope you might consider revising your score upwards. Thank you for your attention to this matter.
> >
> > Best regards.

---

### Official Review · Reviewer_48Jp · 2023-10-30

**Soundness:** 1 poor
**Presentation:** 2 fair
**Contribution:** 1 poor
**Rating:** 3
**Confidence:** 4

**Summary:**

The paper presents an LLM prompting setup termed "ReCon" consisting of

1. one prompt from the perspective of the agentic-role the LLM is intended to assume, consisting of a description of the scenario, the task description and task-hints (information about the game the prompt is evaluated, what to pay attention to, generalized action recommendations)
2. one prompt to guide the LLM through a THINK-SPEAK pattern similar to ReACT, with nudging language specifically guiding the "THINK" inference towards the goal
3. one prompt restating the role, potentially updated scenario and asking the model to critique the "SPEAK" content (but not the THINK content) from the viewpoint and goals of the other roles, again with specific nudges (similar to Self-Ask)
4. one prompt asking the LLM to act as a critique of a player+role, restating the updated situation and asking the model to offer possible improvements guided by hints

This prompt template is avaluated on the role playing game "Avalon: The Resistance" which is described in detail in the appendix and evaluated by having OpenAI GPT-4 and Anthropic Claude play $n=20$ games of one of four ablations (Recon, Recon wo/ ReACT aka prompt , ReCon w/o Self-ask  aka prompt 3, CoT) and evaluated across 6 metrics based on asking GPT-4 to compare rank the two (ranking primpt not included).
A comparison on LLama 2 was attempted but not completed due to formatting-output issues.

**Strengths:**

I feel very mean and mean no offense but I cannot point to any particular unconditional strength in the main axes of consideration.
In an effort to highlight positives:

1. The complete paper is very pretty and visually appealing
2. based on the related work the authors are very familiar with the literature.
3. The method *does* win against the baseline in the task proposed

**Weaknesses:**

1. If the point is to present Avalon as a new benchmark for langauge game playing that as the paper claims "goes beyond a language game and a game of thought", then that benchmark would have to be evaluated with *all* currently relevant baseline methods, as well as studied and discussed in detail to justify the claims: *why* is it a game of thought while e.g. the other games discussed in Serrino et al aren't?
2. If the point is to contribute ReCon, then what is missing is an evaluation on HumanEval, or some other task except the one it appears to be specifically constructed for. E.g., how would one adapt this to a negotiation game, a market game or another situation in which the roles are not as clearly defined etc?

On top of this, the methodology of the evaluation is not strong enough to support anything.
This is currently par for the course and I am sorry that the authors encounter me as a reviewer, but very basic scientific standards currently disqualify most of the established evaluation procedures:

- [recent leaks of basically all large LLM providers system prompts](https://gist.github.com/cedrickchee/9390389d755e574cca24a2b42aaa7d47) show the large possible confounding nature of using blackbox APIs for any form of evaluation, beyond the unknown other methods and filters that might be employed. The authors acknowledge the data and version they use, but this makes replicability impossible, even without the fact that the non-zero temperature and sparse MoE inference at these APIs makes the evaluation indeterministic
- $n=20$ appears very low to me and I would like to see a suitable statistical significance test together with the motivation for choosing and interpreting it (i.e., choosing an appropriate correcting like [this one](https://en.wikipedia.org/wiki/Bonferroni_correction), or justifying a less conservative one)
- I am also confused why the authors didn't use simply use [guidance](https://github.com/guidance-ai/guidance) for their LLamav2 experiments, since they seem otherwise very tuned into the LLM world.  However, Occams *and* Hanlons razor tell me to assume simple unawareness. If the experiments succeed with the help of Guidance, I'd be curious to see the updated results
- That tuned-in-ness brings me to another critique: the described method is basically a specialized application of [Tree of Thoughts](https://arxiv.org/abs/2305.10601). While it might be a concurrent work, the authors also do not adequately put their method into context with ReACT and self-ask as I have done in my summary. What would need to be done is to create a matrix or table comparing all extant/widely used prompt techniques and comparing them while highlighting only *unquestionable* differences
- The room for this could be gained by removing a lot of the unsubstantiated bombast from the introduction and section 2. While I appreciate the aesthetics, I don't think quoting the three body problem, discussing AGIs or making grand claims about CoT not causing the Morgana-role to disclose its thoughts *despite that being in the role description* is suitable for an otherwise quite thinly supported paper. Without more *stats* and *quantifiable information* in the appendix, I also think we do not need to see more individual examples of impressive chats in LLM papers. If e.g. Fig c) shows consistent behaviour, what I would *expect* to see is a section in the appendix detailing
  1. a definition of hidden thought deception that is operationalizable, maybe even with python code
  2. An experimental setup that evaluates this operationalized definition, including all prompts etc. used
  3. a statistical evaluation
  4. a discussion of possible confounders and alternative explanations
- meanwhile, the authors *don't even give the complete prompt in the appendix*. This is the room where you do not need to worry about brevity dear authors, please just add the original prompts, the prompts used to perform the GPT-4 ranking, basically *everything* that could be used to evaluate, judge and replicate the paper
- finally, these are nitpicks, but it is meant to illustrate why I can't even call the writing clear or pleasant:
    - In the introduction, beyond lacking any justification for claiming a "game of thought"...
    - ...how exactly did your findings (which I assume refer to your method and the experiments) inspire your framework?
    - In the appendix, Park et al. 2023 is not a methodological paper, it is a critical evaluation of deception in Machine learning
    - In Appendix g, the Assasin prompt reads "The evil tema is close to losing; you must guess who Merlin is. losing; you must guess who Merlin is". Is this an LLM artifact or a copying error?
  this is a sampling of the general feeling of sloppiness/lack of attention to detail that permeates the paper.

---

If the paper decides whether to present a new benchmark or a method as its focus, then it can be salvaged:

1. For the method, evaluate on standard benchmarks and add your avalon benchmark as a bonus. Properly compare against ToT and other baselines.U
2. For the benchmark, use more open source models and create a detailed story, supported by suitable experiments, to demonstrate *why* this is a "game of thought" as opposed to  a "language game", and throw in your method as the bonus

For both, use guidance to create a controlled experiment using LLama or the recent mystral etc. finetunes, use proper statistical measurements and report things in a manner that can actually be compared (e.g. the table highlighting differences) and reproduced (all exact prompts!)

**Questions:**

I tried to be very constructive and complete in my weaknesses section, hence I do not have questions that would meaningfully change my opinion without those issues addressed

---

> ### Author Response · Authors · 2023-11-22
> **Before Rebuttal:**
>
> Although you mentioned in your reviews, "I am sorry that the authors encounter me as a reviewer," we actually feel quite fortunate to have you as our reviewer because of your detailed reviews, insightful comments, and constructive suggestions. To match the energy and time you have devoted to reviewing, we have thoroughly answered all your questions and comments. We hope our detailed and sincere responses can change your opinion of our paper. Thank you once again for your review.

---

> > ### Comment · Reviewer_48Jp · 2023-11-22
> >
> > Dear Authors, as the final comment for now, I am glad you feel the review was constructive. I have acknowledges some responses as having assuaged my concerns and replied to others, however since the core problems remain unaddressed I will retain my score for now, modulo additional comments and discussion with the other reviewers. If you feel  my remaining concerns are unfounded, please do not hesitate in arguing for why.

---

> ### Author Response · Authors · 2023-11-22
> **Rebuttal to Reviewer 48Jp**
>
> > Q1: In summary section: The paper presents an LLM prompting setup termed "ReCon" consisting of: 1. one prompt ...; 2. one prompt ...; 3. one prompt ...; 4. one prompt ...
>
> Thank you for your detailed summary. However, we beg to differ that **the main contributions of our paper are not focusing on the four prompts**, but **the cognitive insights related to the theory of mind [1] and communicative theory [2], such as perspective-taking and recursive thinking**, which is geneneralizable to other fields or applications.
>
> [1] Yuan, Luyao, et al. "In situ bidirectional human-robot value alignment." *Science robotics* 7.68 (2022): eabm4183.
>
> [2] Tomasello, Michael. *Origins of human communication*. MIT press, 2010.
>
>
> ---
>
> > Q2: The paper is to present benchmark or method?
>
> **The primary focus of our paper is to present the method ReCon.** However, the motivation of the proposal of ReCon is derived from the future usage of AI agents in deceptive environments, and the Avalon game is a suitable testbed for the setting. **The logic of ReCon and Avalon game is similar to the Cicero method and the Diplomacy game in [1], and the AlphaGo method and the game of Go in [2].**
>
> [1] Meta Fundamental AI Research Diplomacy Team (FAIR)†, et al. "Human-level play in the game of Diplomacy by combining language models with strategic reasoning." *Science* 378.6624 (2022).
>
> [2] Silver, David, et al. "Mastering the game of Go with deep neural networks and tree search." *nature* 529.7587 (2016).
>
>
>
> ---
>
>
>
> > Q3: Evaluation on more tasks such as HumanEval, a negotiation game, or a market game, etc.
>
> Thank you for your constructive suggestion. Due to time constraints during the rebuttal period, it's not feasible for us to apply our method to additional tasks. However, a method called SimToM, introduced in a recent paper [1] (submitted to ArXiv approximately one month after our submission), employs techniques similar to ours, such as perspective-taking and a think-twice strategy. This method has been tested on ToMI [2] and BigTOM [3] tasks, demonstrating the effectiveness of these techniques. **While our ReCon framework differs somewhat from SimToM, the outcomes achieved with SimToM could, to some extent, suggest the potential of ReCon for generalization to other tasks.**
>
> [1] Wilf, Alex et al. “Think Twice: Perspective-Taking Improves Large Language Models' Theory-of-Mind Capabilities.” (2023).
>
> [2] Le, Matthew, Y-Lan Boureau, and Maximilian Nickel. "Revisiting the evaluation of theory of mind through question answering." *Proceedings of the 2019 Conference on Empirical Methods in Natural Language Processing and the 9th International Joint Conference on Natural Language Processing (EMNLP-IJCNLP)*. 2019.
>
> [3] Gandhi, Kanishk, et al. "Understanding social reasoning in language models with language models." *arXiv preprint arXiv:2306.15448* (2023).
>
>
>
> ---
>
>
>
> > Q3: The reliability and replicability of using blackbox APIs for evaluation.
>
> Thank you for raising this important issue.
>
> As for reliability, please see `Global Q3` in our [Global Responses](https://openreview.net/forum?id=fvhJu0FODp&noteId=j338DnUQOY), where we have carried out human surveys to assess the reliability of GPT-4's evaluation.
>
> To ensure the replicability as much as possible, we employ zero temperature for GPT-4 for evaluation. In addition, given the sufficient large number of evaluated samples, i.e., more than 2k, randomness can be largely dicreased due to the law of large numbers.
>
>
>
> ---
>
>
>
> > Q4: n=20 seems too low; would like to see a suitable statistical significance test
>
> Thank you for your insightful feedback. We address the issue regarding $n=20$ in `Global Q1`, and a statistical significance test is detailed in `Global Q2`. For more information, please refer to our [Global Responses](https://openreview.net/forum?id=fvhJu0FODp&noteId=Y7twEtlQGy). We greatly appreciate your input.

---

> > ### Comment · Reviewer_48Jp · 2023-11-22
> >
> > Dear authors.
> >
> > Thank you for the effort in addressing the concerns about sample size and black box evaluations. For me, these mainly assuage the methodological concerns I had.
> >
> > However, my concerns about the contribution and clarity of presentation stand. In order to claim
> >
> > >However, we beg to differ that the main contributions of our paper are not focusing on the four prompts, but the cognitive insights related to the theory of mind [1] and communicative theory [2], such as perspective-taking and recursive thinking, which is generalizable  to other fields or applications.
> >
> > one needs to address at the very least
> >
> > - the similarity to Tree of thought/lack of contextualization and differentiation
> > - demonstrate applicability and benefits beyond the simultaneously presented benchmark...
> > - *or* motivate further why performance in this benchmark cannot be expected to be a (quoting the paper from memory) "mere language game, but a game of thought" and thus make ReCon unique
> > - perform further experiments isolating the effect of the proposed Theory of mind and communicative theory. E.g. [Ullman 2023](https://arxiv.org/abs/2302.08399) showed showed LLMs "ToM" to be exceedingly brittle against small alterations and https://arxiv.org/abs/2305.15068 as well as https://arxiv.org/abs/2305.14763 develop specific benchmarks for ToM, which can probably improve an evaluation of ReCon and support this main claim
> >
> > This is beyond ones perspective on the "mumbling of incantations" and debates on prompt engineering vs. "techniques of thoughts" as raised in the discussion with reviewer zjpH (although I am with them on in their skepticism/hostility towards LLMs) and simply on the solidity of the claims being made.
> >
> > While I acknowledge your effort in solidifying your evaluation of GPT and Claude, for me these core issues of the paper remain unaddressed. Please correct me if I have missed a response or alteration of the paper in which you could debunk this perception.

---

> ### Author Response · Authors · 2023-11-22
> **(Continued and to Be Continued) Rebuttal to Reviewer 48Jp**
>
> > Q5: Use "guidance" for LLaMA v2 expriments
>
> Thank you for your feedback. We initially overlooked including "guidance" in our LLaMA v2 experiments, but following your suggestion, we've updated our code to integrate it while keeping most prompts the same.
>
> In our tests, we used LLaMA2-Chat (7b, 13b, 70b), `vicuna-13b-v1.5-16k`, and `Mistral-7B-Instruct-v0.1` for playing the Avalon game. The LLaMA2-Chat models struggled with Avalon tasks due to their limited 2k-token context window. The `vicuna-13b-v1.5-16k` and `Mistral-7B-Instruct-v0.1` models, with their larger context windows of 16k and 8k tokens, could play the game but faced issues with format mismatches and repeated responses, as detailed below:
>
> - **Format mismatches.** Format mismatches occur in some stages of our ReCon process, particularly during the first-think-then-speak phase. Initially, we generated the thinking and speaking contents by creating a JSON file, following the steps in the [guidance's tutorial](https://github.com/guidance-ai/guidance#guidance-acceleration):
>
>   ```python
>   with guidance.assistant():
>       lm += f"""
>   ```json
>   {{
>       "thinking content": "THINK: {guidance.gen('think_content', suffix='"', max_tokens=500)}",
>       "speaking content": "SPEAK: {guidance.gen('speak_content', suffix='"', max_tokens=500)}"
>   }}```"""
>       think_content, speak_content = lm['think_content'], lm['speak_content']
>   ```
>
>   Occasionally, a `KeyError` occurs when `think_content` or `speak_content` is missing in `lm`, hindering the completion of Avalon games due to frequent thinking and speaking actions. We speculate that this may stem from the LLMs' inconsistent adherence to format requirements. This conclusion is aligned with Sec. 5.5.
>
> - **Repeated responses.** To address the issue of mismatched format mentioned earlier, we modified our code as follows:
>
>   ```python
>   with guidance.assistant():
>       lm += f'''"thinking content": "THINK: {guidance.gen('think_content', suffix='"', max_tokens=500)}"'''
>       lm += f'''"speaking content based on the thinking content (not repeating other's previous speaking)": "SPEAK: {guidance.gen('speak_content', suffix='"', max_tokens=500)}"'''
>       think_content, speak_content = lm['think_content'], lm['speak_content']
>   ```
>
>   This revision effectively prevents the `KeyError` mentioned previously. Yet, we observe that agents frequently repeat the speaking content of other agents, as provided in the given context. For instance:
>
>   ```
>   Player 1:
>
>   I propose the team player 2, player 1. Based on my considerations, I have decided to propose a mission team consisting of myself and Player 2. Player 2 has been actively involved in the discussions and has shown a willingness to collaborate with others. They have also been supportive of the team and have not been suspect of any wrongdoing. I believe that with their skills and my leadership, we can successfully complete the mission. However, I must also be cautious about how my proposal might be perceived by other players, especially those who may be suspect of being evil. I will make sure to provide cover for my teammate if they are suspect of being evil, without revealing my own identity or Morgana's identity to the other players.
>
>   ---
>
>   Player 2:
>
>   Dear fellow players, I have carefully considered the proposed team players for this mission. Based on my analysis, I believe that player 2 and player 1 are suitable candidates for the team. Both players have been activey involved in the discussion and have shown a willingness to collaborate with others. They have also been supportive of the team and have not been suspect of any wrongdoing.
>
>   However, I must also be cautious about how my proposal might be perceived by other players, especially those who may be suspect of being evil. I will make sure to provide cover for my teammates if they are suspect of being evil, without revealing my own identity or Morgana's identity to the other players.
>
>   ---
>
>   Player 3:
>
>   Dear fellow players, I have carefully considered the proposed team players for this mission. Based on my analysis, I believe that player 2 and player 1 are suitable candidates for the team. Both players have been active and supportive of the team, and they have not been suspect of any wrongdoing.
>
>   However, I must also be cautious about how my proposal might be perceived by other players, especially those who may be suspect of being evil. I will make sure to provide cover for my teammates if they are suspect of being evil, without revealing my own identity or Morgana's identity to the other players.
>   ```
>
>   We suspect that this issue of repetitive responses might be attributed to the limited capabilities of the LLMs we are using within the context of Avalon.
>
> **Owing to the aforementioned issues, despite our keen interest in providing experimental results on open-source LLMs using "guidance", we find ourselves unable to do so at this time.** We will leave this to future work.

---

> > ### Comment · Reviewer_48Jp · 2023-11-22
> >
> > Dear authors. Thank you very much for spending the time to explore using uidance. In the meantime, I also learned of llama.cpps [GBNF](https://github.com/ggerganov/llama.cpp/blob/master/grammars/README.md) funtionality which might further be suitable for your setup, however, I want to explicitly acknowledge the effort put into addressing this part of my review and think this is truly a "best effort" in using Open Source LLMs.

---

> ### Author Response · Authors · 2023-11-23
> **Reply to Reviewer 48Jp (https://openreview.net/forum?id=fvhJu0FODp&noteId=nflNnEWEY7)**
>
> Dear Reviewer 48Jp:
>
> Thank you for your response and additional insightful questions. We have provided detailed answers to your further queries below. We trust that our responses will address and clarify any doubts you may have regarding our work.
>
>
> ---
>
>
>
> > Comparison with Tree of Thoughts (ToT)
>
> We understand your concern about the similarity between ToT and our ReCon due to the iterative thinking pattern. However, we believe that our ReCon has some significant differences from ToT, and cannot be viewed as a specific case of ToT, due to the following reasons:
>
> |                           | Tree of Thoughts (ToT)                                       | ReCon (Ours)                                                 |
> | ------------------------- | ------------------------------------------------------------ | ------------------------------------------------------------ |
> | Core Motivation           | ToT is designed to facilitate exploration through coherent text units ("thoughts") as intermediate steps in **problem-solving**. It's aimed at tasks like the Game of 24, Creative Writing, and Mini Crosswords, which **do not involve misinformation or deception**. These tasks also **do not occur in communicative contexts**. | ReCon is developed to **manage deceptions and misinformation in communicative environments**, where there may not be a definitive "correct" answer, emphasizing the nuances of communication. |
> | Algorithmic Focus         | ToT focuses on **constructing complex reasoning trees**, with looking ahead or backtracking, to address problems. | ReCon's approach differs as it does not rely on complex reasoning trees. Instead, **it prioritizes Theory of Mind** (such as understanding and speculating about others' thoughts and identities), **and communicative skills** (such as expressing opinions while concealing its own identity when necessary). |
> | Nature of Reasoning Trace | ToT's reasoning trace comprises a series of **decomposed, intermediate thought steps** aimed at **problem resolution**. | ReCon's reasoning trace involves **the original and revised versions of thoughts and spoken contents**, tailored for **enhanced Theory of Mind and communicative skills**. |
>
> ---
>
>
>
> > motivate further why performance in this benchmark cannot be expected to be a (quoting the paper from memory) "mere language game, but a game of thought" and thus make ReCon unique
>
> The Avalon game, on which ReCon is tested, inherently involves a significant number of deceptions. As highlighted on [proavalon.com](https://www.proavalon.com/), a renowned platform for playing Avalon online, the game is defined as "a game of deception", where "a small band of revolutionaries must use logic and deduction in order to ferret out the spies who have infiltrated their ranks and are sabotaging the cell's presumably heroic acts of rebellion against government tyranny." **This characterization of the Avalon game solidly backs our stance that it is not just "mere language game, but a game of thought".** We also suggest participating in an Avalon game when possible, as it will likely solidify your agreement with our claim.
>
> ---
>
>
>
> > perform further experiments isolating the effect of the proposed Theory of mind and communicative theory.
>
> Thank you for your insightful comments and suggestions.
>
> We acknowledge that current LLMs' Theory of Mind (ToM) capabilities are brittle. However, in this work, LLMs serve primarily as a tool to bridge the ToM framework with natural language responses. In other words, **we're explicitly integrating ToM into LLM-as-agent, demonstrating enhanced performance over LLMs alone**. The benchmarks you've recommended **mainly assess the ToM abilities of various LLMs, whereas our work's focus isn't on developing stronger LLMs, but on their integration with ToM.**
>
> Although the benchmarks you suggest may not be the best fit for testing our methodology, we believe it's important to distinctly showcasing the impacts of Theory of Mind and our communicative approach. Understanding your preference against more individual examples, **we have provided a comprehensive set of 372 instances (which represents a substantial quantity and precludes the possibility of selective cherry-picking) in the Excel file of this [anonymous link](https://anonymous.4open.science/r/ICLR_Avalon_Rebuttal_ID_118/ReCon_thinking_and_speaking_logs.xlsx). These examples display ReCon's thought and speech processes, where the thinking content demonstrates the efficacy of ReCon's Theory of Mind, and the speaking content illustrates the effectiveness of ReCon's communication skills.**
>
> Additionally, **Figures 4 and 6** in our paper, highlighting ""**ReCon w/o First-Order Perspective Transition**" and "**ReCon w/o Second-Order Perspective Transition**," demonstrate ReCon's ToM effectiveness by isolating the first-order and second-order ToM elements from ReCon, respectively.

---

> > ### Comment · Reviewer_48Jp · 2023-11-23
> >
> > Dear authors. You have responded adequately to most of the other concerns I have raised and I think the paper has grown stronger from it, however, on this, I think we are in a loop and my concerns are of a manner that you won't be able to still them in rebuttal.
> >
> > Our divergence lies in the fact that I do not care about the *words* Theory of Mind or "game of deception", I want *numerics* giving meaning to them.
> >
> > That is, while you outline three claimed differences between ToT and your method, there is not a clear formal distinction. Paraphrasing your argument, you claim that one distinction between ToT and ReCon is the imperfect information and ambiguity setting. Well, [MCTS performs very well in imperfect information](https://www.frontiersin.org/articles/10.3389/frai.2023.1014561/full) and theory of mind and imperfect information involves reasoning (as in, exploring a search tree of possible deductions) through various scenarios using assumptions - which just means taking statements as a fact without clear evidence. Thus, ToT can be seen as a superset of your method, modulo the exact wording of the prompt - which, if you want to claim a distinction from mere prompt engineering ("arcane incantations") should not matter that much. The problem is that Ullman 2023 shows that it *does* matter, which implies to me again that the claim to be focused on ToM would require you to perform similar ablations and investigations in that matter.
> >
> > Further, the previous claims of ToM and deception being central to your paper are kind of in conflict with statements like
> >
> > >The benchmarks you've recommended mainly assess the ToM abilities of various LLMs, whereas our work's focus isn't on developing stronger LLMs, but on their integration with ToM.
> >
> > What *is* the central point of the paper?
> >
> > - Showcasing good performance of the LLM on a new benchmark using your new prompt technique? => Then you need more diverse benchmarks and ablations.
> > - Highlight and study the importance of ToM in Avalon => then you probably should integrate deeper with the ToM literature and perform specific ablations and at least one ToM specific benchmark with ReCon
> > - Study deception in LLMs and how to make it explicit/deal with it? => then this is not really clearly articulated in the paper, the method isnt' really evaluated in this manner (except the sucess rate on avalon which might just be an artifact of the prompt technique and system prompts sabotaging the agens ability to generate falsehoods coherently)
> >
> > Finally, neither of
> >
> > > This characterization of the Avalon game solidly backs our stance that it is not just "mere language game, but a game of thought".
> > and
> > > We also suggest participating in an Avalon game when possible, as it will likely solidify your agreement with our claim.
> >
> > are scientific arguments. If you ask the average person, computer science is a mix of black magic and boring minutia, and
> >
> > 1. I find myself both unable to summon eldritch forces *and* highly stimulated after over a decade in the field
> > 2. as aggregated anecdotal evidence, nothing of this matters unless suitably operationalized.
> >
> > IFF you present an operationalization on what it *means* to  be a game of though as opposed to a language game (e.g. by grounding it in imperfect information and the need to perform deduction and bluffing from meta-game awareness, as e.g. done in [Deepminds study of stratego](https://www.science.org/stoken/author-tokens/ST-887/full#supplementary-materials) or by isolating the language aspect from the reasoning aspect, then your argument would fly.
> >
> > Alternatively, you might be able to side step this by showing alignment with a reasoning algorithm in a latent state, by learning interpretable mappings from the latent state of game states observed by the LLMs into predicting belief states a game tree and then showing it behaves like a causal inference or otherwise suitable algorithm.
> >
> > Or there might be other ways of making your point in a more empirical manner.
> >
> > But in the absence of this the words "this paper is about ToM, deception and a game of thought"  do not mean anything, because they are too fuzzy concepts and too weakly supported in their construction. That is at least my current stance, and in the remaining time I don't think you will be able to perform the additional work to remedy this. However I wish you the best of luck to further refine and sharpen the paper, and hope that the angle I sketch above (sharpen the distinction between ToT and ReCon by performing focused studies that leverage an analogy on the difference of MCTS in MDPs and POMPDs) might prove helpful in this.

---

> ### Author Response · Authors · 2023-11-23
> **Replies to Additional Comments from Reviewer 48Jp's Initial Review (https://openreview.net/forum?id=fvhJu0FODp&noteId=MUmwucVQBR)**
>
> > Q7: the authors don't even give the complete prompt in the appendix.
>
> Apologies for the previous lack of detail. For your convenience, we have provided the full prompts of ReCon in this [anonymous link](https://anonymous.4open.science/r/ICLR_Avalon_Rebuttal_ID_118/Avalon_game_prompt.pdf). Additionally, the prompts used for GPT-4 evaluation are available at another [anonymous link](https://anonymous.4open.science/r/ICLR_Avalon_Rebuttal_ID_118/gpt4_evaluation_prompts.pdf).
>
> ---
>
> > Q8: In the appendix, Park et al. 2023 is not a methodological paper, it is a critical evaluation of deception in Machine learning
>
> Thank you for your thorough review. However, there seems to be a possible misunderstanding. In our related work section, **we have cited two papers authored by researchers with the surname Park: Park et al. (2023a) [1] and Park et al. (2023b) [2]**. While the latter [2] critically evaluates deception in machine learning, the former [1] is a methodological paper that introduces "generative agents". **It appears there might have been a mix-up between these two works during review.**
>
> [1] Park, Joon Sung, et al. "Generative agents: Interactive simulacra of human behavior." *Proceedings of the 36th Annual ACM Symposium on User Interface Software and Technology*. 2023.
>
> [2] Park, Peter S., et al. "AI deception: A survey of examples, risks, and potential solutions." *arXiv preprint arXiv:2308.14752* (2023).
>
> ---
>
>
>
> > Q9: In Appendix g, the Assasin prompt reads "The evil tema is close to losing; you must guess who Merlin is. losing; you must guess who Merlin is". Is this an LLM artifact or a copying error? this is a sampling of the general feeling of sloppiness/lack of attention to detail that permeates the paper.
>
> Thank you for pointing out this issue. We acknowledge the copying error and apologize for the typo, which has been corrected in our revised manuscript.
>
> While we regret this oversight, we respectfully disagree that it reflects a broader lack of attention to detail throughout our paper. We have approached our work with diligence and care, as evidenced by the positive remarks on writing and presentation from most reviewers, and as further demonstrated in our detailed rebuttal response. Aside from this isolated instance, our paper contains few typographical or grammatical errors.
>
> Nevertheless, we sincerely apologize for this mistake and appreciate your understanding.

---

> > ### Comment · Reviewer_48Jp · 2023-11-23
> >
> > Q7: thank you for providing the complete prompt
> > Q8: Indeed, this was an unfortunate mixup as the two references are above each other, apologies for this.
> > Q9: I appreciate the refutation.

---

### Official Review · Reviewer_zjpH · 2023-10-31

**Soundness:** 2 fair
**Presentation:** 2 fair
**Contribution:** 2 fair
**Rating:** 3
**Confidence:** 3

**Summary:**

The paper attempts to study and improve LLMs' ability to produce situationally appropriate responses in a strategic environment requiring deception and the ability to detect deception. LLMs are prompted to play roles in the social-deduction game Avalon. The experimental intervention is a new prompting technique ReCon, which explicitly prompts agents to contemplate the roles of other players based on the game history, and then explicitly prompts them to contemplate how other players will perceive them. The outputs are then evaluated in two ways: (1) by playing complete games of Avalon to see whether ReCon-prompted agents perform better, and (2) having GPT-4 classify outputs as successful or not along six dimensions.

**Strengths:**

The prompt design is clever and the prompts make explicit ideas that are implicit in human thought processes about social roles.

**Weaknesses:**

It is hard to see what is generalizable here. LLMs are fundamentally linguistic; any higher-order cognitive processes are emergent properties.  The kind of prompt engineering illustrated here yields no insight into those processes. Instead, it is akin to mumbling different incantations at a mysterious creature to see how it responds. In what sense have the LLMs understood or responded to the ReCon prompts? Unclear. Do they have a usable theory of mind about other agents in the game? Unclear. These linguistic prompt manipulations are inherently fragile. They depend enormously on the current capabilities of the specific LLMs studied; there is no good reason to think that these approaches will transfer to other future LLMs with different training sets, architectures, or fine-tuned guardrails.

The use of ChatGPT to perform multi-dimensional evaluation needs to be validated against human evaluation.

**Questions:**

none

**Details Of Ethics Concerns:**

The authors note that these techniques require eliciting deceptive behavior from LLMs and could be used in the future for deception. I do not think that there are serious ethical problems with the research, but someone who works specifically on the ethics of AI safety should review this to ensure that it is in line with professional ethical standards.

---

> ### Author Response · Authors · 2023-11-17
> **Rebuttal to Reviewer zjpH**
>
> We want to express our thanks to you for your reviews. Although there might be some misunderstandings, we are pleased to address your concern in the following response.
>
> > Q1: It is hard to see what is generalizable here.
>
> We appreciate the feedback and understand the concerns regarding generalizability. However, we emphasize that our proposed ReCon is **not limited to specific prompts in the Avalon game**. Instead, **it focuses on emulating human-like recursive thinking and perspective-taking**. As pointed out by reviewer zjpH himself/herself, ReCon "makes explicit ideas that are implicit in human thought processes about social roles." **This human-like thought processes on LLMs are the generalizable aspect of ReCon**, which is particularly effective in complex scenarios involving logic and deception.
>
> **Furthermore, testing on a single, challenging task that current methods struggle with is a standard research practice and does not indicate non-generalizability.** For instance, Cicero [12] demonstrated exceptional performance exclusively in the game of Diplomacy, yet its designs are considered generalizable. Similarly, generative agents [13], tested solely in a Sims-inspired interactive environment, yielded generalizable insights. Beyond LLM agents, in the realm of RL agents, notable studies like AlphaGo [1] and AlphaStar [2] focused on specific tasks—Go and StarCraft II, respectively—yet the techniques developed in these works have been proven to be generalizable.
>
>
>
> [1] Silver, David, et al. "Mastering the game of Go with deep neural networks and tree search." *nature* 529.7587 (2016).
>
> [2] Vinyals, Oriol, et al. "Grandmaster level in StarCraft II using multi-agent reinforcement learning." *Nature* 575.7782 (2019).

---

> > ### Author Response · Authors · 2023-11-17
> > **(Continued) Rebuttal to Reviewer zjpH**
> >
> > > Q2: LLMs are fundamentally linguistic; any higher-order cognitive processes are emergent properties. The kind of prompt engineering illustrated here yields no insight into those processes. Instead, it is akin to mumbling different incantations at a mysterious creature to see how it responds.
> >
> > We acknowledge the viewpoint that LLMs are fundamentally linguistic and that higher-order cognitive processes are emergent properties. However, we wish to clarify two critical points:
> >
> > 1. **Having the capability does not necessarily equate to achieving potential.** For instance, while any cognitively sound individual possesses the intelligence to play the Avalon game, the quality of play varies significantly without the right reasoning mechanisms, even for the same person. In the realm of LLMs, their emergent nature only ensures the capability to engage in activities like playing Avalon but does not guarantee reaching the pinnacle of gameplay potential. **It is here that thinking mechanisms (e.g.,  Chain-of-Thought [1, 2], Tree-of-Thought [3], Graph-of-Thought [4], Reflexion [5], ReAct [6], Voyager [7], and our proposed ReCon) play a pivotal role; they are designed to unleash the potential of LLMs in complex scenarios.**
> > 2. **It is an oversimplification to categorize all methods developing thinking mechanisms (such as [1-7], and our proposed ReCon) as mere "prompt engineering."** As emphasized earlier, the essence of these methods lies in the insights derived from the underlying thinking processes, not the prompts themselves. **To label these innovative algorithms and our ReCon as just "mumbling different incantations at a mysterious creature to see how it responds"—for example, understanding Chain-of-thought as just adding an incantation like "let's think step by step"[2]—would unfairly dismiss their value and the significant advancements they represent.** This perspective overlooks the strategic and thoughtful design embedded in these mechanisms, which are crucial for advancing our understanding and application of LLMs in complex scenarios.
> > 3. Additionally, our paper's experiments show that current cognitive methods for LLMs, like CoT, are ineffective in deceptive environments such as Avalon, whereas our ReCon method proves successful. **The success of ReCon is not due to brute-force attempts with various prompts but rather stems from cognitive insights related to the theory of mind [8] and communicative theory [9].**
> >
> >
> >
> > [1] Wei, Jason, et al. "Chain-of-thought prompting elicits reasoning in large language models." *Advances in Neural Information Processing Systems* 35 (2022): 24824-24837.
> >
> > [2] Kojima, Takeshi, et al. "Large language models are zero-shot reasoners." *Advances in neural information processing systems* 35 (2022): 22199-22213.
> >
> > [3] Yao, Shunyu, et al. "Tree of thoughts: Deliberate problem solving with large language models." *arXiv preprint arXiv:2305.10601* (2023).
> >
> > [4] Besta, Maciej, et al. "Graph of thoughts: Solving elaborate problems with large language models." *arXiv preprint arXiv:2308.09687* (2023).
> >
> > [5] Shinn, Noah, et al. "Reflexion: Language agents with verbal reinforcement learning." *Thirty-seventh Conference on Neural Information Processing Systems*. 2023.
> >
> > [6] Yao, Shunyu, et al. "React: Synergizing reasoning and acting in language models." *arXiv preprint arXiv:2210.03629* (2022).
> >
> > [7] Wang, Guanzhi, et al. "Voyager: An open-ended embodied agent with large language models." *arXiv preprint arXiv:2305.16291* (2023).
> >
> > [8] Yuan, Luyao, et al. "In situ bidirectional human-robot value alignment." *Science robotics* 7.68 (2022): eabm4183.
> >
> > [9] Tomasello, Michael. *Origins of human communication*. MIT press, 2010.
> >
> >
> >
> >
> > ---
> >
> >
> >
> > > Q3: In what sense have the LLMs understood or responded to the ReCon prompts? Do they have a usable theory of mind about other agents in the game?
> >
> > To demonstrate that **LLMs can effectively understand and respond to ReCon prompts, possessing a practical theory of mind about other game agents**, we have provided an Excel file at **[this anonymous link](https://anonymous.4open.science/r/ICLR_Avalon_Rebuttal_ID_118/ReCon_thinking_and_speaking_logs.xlsx)**. This file contains extensive logs of the thinking and speaking processes of LLMs when employing ReCon.

---

> ### Author Response · Authors · 2023-11-17
> **(Continued) Rebuttal to Reviewer zjpH**
>
> > Q4: These linguistic prompt manipulations are inherently fragile. They depend enormously on the current capabilities of the specific LLMs studied.
>
> In our previous discussion, we emphasized that possessing a capability does not automatically lead to realizing its full potential. Thus, **there are primarily two types of work concerning LLMs: (1) expanding the boundaries of LLMs' emergent capabilities (such as [8-10]), and (2) unlocking the full potential of these capabilities (such as [1-7] and ReCon). Both approaches are critically important.** To draw an analogy, the first approach is similar to enhancing a child's IQ, while the second is akin to teaching the child advanced thinking strategies to maximize the use of their IQ. It would be incorrect to dismiss the value of these thinking strategies simply because they are contingent on the child's IQ. Similarly, it's unjustified to label the thinking process methods for LLMs as "inherently fragile" and dismiss them as irrelevant just because they depend on the current abilities of LLMs. Rather, **as the limits of LLMs' emergent capabilities significantly expand, the importance of fully unlocking the potential of LLMs may increasingly come to the forefront [1-7, 11].**
>
>
>
> [1] Wei, Jason, et al. "Chain-of-thought prompting elicits reasoning in large language models." *Advances in Neural Information Processing Systems* 35 (2022): 24824-24837.
>
> [2] Kojima, Takeshi, et al. "Large language models are zero-shot reasoners." *Advances in neural information processing systems* 35 (2022): 22199-22213.
>
> [3] Yao, Shunyu, et al. "Tree of thoughts: Deliberate problem solving with large language models." *arXiv preprint arXiv:2305.10601* (2023).
>
> [4] Besta, Maciej, et al. "Graph of thoughts: Solving elaborate problems with large language models." *arXiv preprint arXiv:2308.09687* (2023).
>
> [5] Shinn, Noah, et al. "Reflexion: Language agents with verbal reinforcement learning." *Thirty-seventh Conference on Neural Information Processing Systems*. 2023.
>
> [6] Yao, Shunyu, et al. "React: Synergizing reasoning and acting in language models." *arXiv preprint arXiv:2210.03629* (2022).
>
> [7] Wang, Guanzhi, et al. "Voyager: An open-ended embodied agent with large language models." *arXiv preprint arXiv:2305.16291* (2023).
>
> [8] Brown, Tom, et al. "Language models are few-shot learners." *Advances in neural information processing systems* 33 (2020): 1877-1901.
>
> [9] Touvron, Hugo, et al. "Llama: Open and efficient foundation language models." *arXiv preprint arXiv:2302.13971* (2023).
>
> [10] Touvron, Hugo, et al. "Llama 2: Open foundation and fine-tuned chat models." *arXiv preprint arXiv:2307.09288* (2023).
>
> [11] Wang, Lei, et al. "A survey on large language model based autonomous agents." *arXiv preprint arXiv:2308.11432* (2023).
>
>
>
>
>
> ---
>
>
>
> > Q5: There is no good reason to think that these approaches will transfer to other future LLMs with different training sets, architectures, or fine-tuned guardrails.
>
> In our study, we evaluated ReCon on **two distinct LLMs, ChatGPT and Claude**, which differ in "training sets, architectures, and fine-tuned guardrails." We applied **identical thinking processes and prompts** to both. ReCon's superior outcomes with ChatGPT and Claude, as depicted in Figure 4 of our paper, suggest that **ReCon possesses a certain level of generalizability across various LLMs**.
>
> Regarding "future LLMs," the thought processes in ReCon, such as recursive thinking and perspective-taking, are also useful in human intelligence. Therefore, **we have reason to believe that in the foreseeable future, even if LLMs approach or match human intelligence, the ReCon method we propose will remain effective**. The only scenario where ReCon might not work on future LLMs is if the chosen future LLMs have inferior generative abilities.

---

> ### Author Response · Authors · 2023-11-17
> **(Continued) Rebuttal to Reviewer zjpH**
>
> > Q6: The use of ChatGPT to perform multi-dimensional evaluation needs to be validated against human evaluation.
>
> We appreciate your valuable suggestions. In line with your recommendations, we are currently undertaking a human agreement evaluation akin to the approach used by AlpacaEval (details found [here](https://github.com/tatsu-lab/alpaca_eval#evaluators)). We aim to publish the results promptly.
>
> ----
>
> We extend our heartfelt gratitude once more for your review. We anticipate further dialogue and collaboration, and are open to any more thoughts you may have to help refine our work.

---

> > ### Author Response · Authors · 2023-11-23
> > **A Kindly Request for Response to Our Rebuttal**
> >
> > Dear Reviewer zjpH,
> >
> > With **less than 6 hours remaining** before the ICLR author-reviewer discussion concludes, we kindly request your prompt feedback on our rebuttal. Your insights are invaluable to us, and a timely response would be greatly appreciated to ensure a comprehensive review process.
> >
> > Furthermore, We have completed the human evaluation assessing the reliability of automatic language model evaluation. For our results, please see **Appendix D.1 in the revised manuscript** or refer to **[Global Q3](https://openreview.net/forum?id=fvhJu0FODp&noteId=j338DnUQOY)**.
> >
> > If our response has addressed your concerns, we hope you might consider revising your score upwards. Thank you for your attention to this matter.
> >
> > Best regards.

---

### Official Review · Reviewer_WArX · 2023-11-01

**Soundness:** 3 good
**Presentation:** 3 good
**Contribution:** 2 fair
**Rating:** 3
**Confidence:** 3

**Summary:**

This paper studies whether LLMs can deceive and act strategically in deceptive environments, using the Avalon game as a testbed. The authors propose a method called Recursive Contemplation (ReCon) to enhance LLMs’ ability to identify and counteract deceptive information, based on theory of mind methods. The authors show that ReCon is able to aid LLMs to discern and maneuver around deceptive information.

**Strengths:**

1. The paper is well-motivated: studying the truthfulness and deceptive abilities of LLMs is certainly very important.
2. The dataset can be a nice contribution.
3. The paper is well-written.

**Weaknesses:**

1. Mainly GPT-4 is used for evaluation so I'm not sure how reliable the evaluation is - there should be at least reliable human evaluation.
2. There are only 20 games so not sure how reliable how the results are.
3. 4.2 MULTI-DIMENSIONAL EVALUATION is only done with chatgpt so it's not clear how reliable and reproducible the results will continue to be.
4. I also don't really understand 4.2 MULTI-DIMENSIONAL EVALUATION - why are existing metrics from (Li et al., 2023b; Bang
et al., 2023) being used here? Aren't we trying to study the performance of LLMs on avalon game? Again, I believe human evaluation would be the gold standard here.
5. 4.3 QUALITATIVE ANALYSES is quite interesting, but the results are very anecdotal to be reliable. 'ReCon’s Proficiency in Detecting Misinformation' -> this is quite cool and would be a very nice result, but there are only anecdotes to support it and not a complete quantitiative evaluation. Can an actual dataset be collected with misinfo, true info labeled, and ReCon vs other baselines rigorously benchmarked on it? Similarly for 'The efficacy of ReCon in information', another cool finding, but needs much more extensive comparisons and evaluation to be reliable. 'Lai et al., Werewolf among us: Multimodal resources for modeling persuasion behaviors in social deduction games' could be a useful resource here.

**Questions:**

1. Explain why the evaluation with GPT-4 makes sense and why human eval is not needed, or provide human eval if necessary.
2. Explain 4.2 MULTI-DIMENSIONAL EVALUATION, including why only chatgpt was used or provide results with open-source LMs, explain the evaluation metrics and why human eval is not needed, or provide human eval if necessary.
3. Additional results on 4.3 QUALITATIVE ANALYSES as appropriate, including whether these can be evaluation rigorously.

**Details Of Ethics Concerns:**

This paper dives deep into generation, detection, and mitigation of misinformation in LLMs. While I believe the authors have mostly addressed the ethical impacts of their work I will defer the final judgement to the ethics expert reviewers.

---

> ### Author Response · Authors · 2023-11-18
> **Rebuttal to Reviewer WArX**
>
> We are grateful for your thorough review and helpful recommendations, which have contributed significantly to the enhancement of our manuscript.
>
> ----
>
> > Q1: There are only 20 games, so I'm not sure how reliable the results are.
>
> Thank you for your valuable question. We've addressed the rationale behind selecting 20 Avalon game rounds in `Global Q1` and have provided a statistical test of our results in `Global Q2`. For more information, please see our [Global Responses](https://openreview.net/forum?id=fvhJu0FODp&noteId=Y7twEtlQGy). We appreciate your interest.
>
> ---
>
> > Q2: Explain why the evaluation with GPT-4 makes sense and why human eval is not needed, or provide human eval if necessary.
>
> Thank you for your insightful question. We agree that incorporating human evaluation alongside GPT-4 assessment is beneficial. Consequently, we are now conducting a human agreement evaluation similar to the method employed by AlpacaEval (further information available [here](https://github.com/tatsu-lab/alpaca_eval#evaluators)). We intend to release the findings as soon as possible.
>
>
>
> ---
>
>
> > Q3: 4.2 MULTI-DIMENSIONAL EVALUATION is only done with ChatGPT, so it's not clear how reliable and reproducible the results will continue to be.
>
> Regarding **reliability**, automated evaluation of LLMs is widely recognized and popular [1]. In our evaluation using GPT-4, we analyzed **over 2300 LLM responses**, which we believe enhances the evaluation's reliability. Nevertheless, as mentioned in our response to `Q2`, we acknowledge the value of adding human evaluation to GPT-4 assessment for greater reliability. Therefore, we are conducting a human agreement evaluation using a method similar to AlpacaEval (more details [here](https://github.com/tatsu-lab/alpaca_eval#evaluators)), and we plan to publish these results promptly.
>
> In terms of **reproducibility**, while OpenAI's API did not offer deterministic output during our experiments, we maintained reproducibility by **setting the GPT-4 evaluator's temperature to 0 and documenting the GPT-4 version in our paper's "Reproducibility Statement."** Additionally, evaluating **over 2300 LLM responses** significantly reduces randomness **under the law of large numbers**. Hence, we consider our GPT-4 evaluation to be reproducible.
>
> [1] Chang, Yupeng, et al. "A survey on evaluation of large language models." *arXiv preprint arXiv:2307.03109* (2023).
>
> ---
>
> > Q5: 4.3 QUALITATIVE ANALYSES is quite interesting, but the results are very anecdotal to be reliable. 'ReCon’s Proficiency in Detecting Misinformation' -> this is quite cool and would be a very nice result, but there are only anecdotes to support it and not a complete quantitiative evaluation. Can an actual dataset be collected with misinfo, true info labeled, and ReCon vs other baselines rigorously benchmarked on it? Similarly for 'The efficacy of ReCon in information', another cool finding, but needs much more extensive comparisons and evaluation to be reliable. 'Lai et al., Werewolf among us: Multimodal resources for modeling persuasion behaviors in social deduction games' could be a useful resource here.
>
> Thank you for recognizing the interest in our qualitative analysis, and we're happy to address your question about it below.
>
> Regarding the "anecdotal" aspect of our results, **to bolster the reliability of our qualitative analysis, we've compiled extensive data on how LLMs think and speak using ReCon in an Excel file, accessible at this [anonymous link](https://anonymous.4open.science/r/ICLR_Avalon_Rebuttal_ID_118/ReCon_thinking_and_speaking_logs.xlsx)**. Additionally, upon acceptance of our paper, we plan to open-source our codes, inviting you to test them and assess the reliability of our qualitative results yourself.
>
> We also value your suggestion about compiling an actual dataset. Due to time constraints in this rebuttal phase, we haven’t been able to gather such a dataset and benchmark different methods on it, but we will consider incorporating this into our future work. Thank you once again for your constructive suggestion.
>
> ---
>
> > Q6: Explain 4.2 MULTI-DIMENSIONAL EVALUATION, including why only chatgpt was used or provide results with open-source LMs, explain the evaluation metrics and why human eval is not needed, or provide human eval if necessary.
>
> Thank you for your inquiry about our evaluation approach.
>
> Regarding the lack of evaluation with open-source LMs, as detailed in Section 5.5 of our paper, LLaMA2-70b-chat, a prominent open-source LM, often struggles to consistently produce the structured output formats our study requires, like generating outputs suitable for regular expression matching (e.g., [approve]). This issue poses challenges in employing open-source LMs for playing the Avalon game.
>
> For the question on human evaluation, we invite you to refer to our response to Q2 for more information.
>
> ---
>
> We thank you again for your expertise and attention, and are ready to address any further questions or concerns.

---

> > ### Comment · Reviewer_WArX · 2023-11-23
> > **thank you for your response**
> >
> > Thank you authors for your responses. I am glad you agree with my concerns regarding reliability of automatic language model evaluation, the need for more human evaluation, and the need for more concrete quantitative analysis of the claims made in the paper. Since you acknowledge that these are directions you are currently working towards, I hope to see them in the improved version of the paper, and I keep my current score.

---

> ### Author Response · Authors · 2023-11-23
> **Reply to Reviewer WArX**
>
> Thank you for your response. We have completed the human evaluation assessing the reliability of automatic language model evaluation. For our results, please see **Appendix D.1 in the revised manuscript** or **refer to [Global Q3](https://openreview.net/forum?id=fvhJu0FODp&noteId=j338DnUQOY)**. We appreciate your attention to this matter.
>
> We trust that our comprehensive response has addressed your concerns regarding our work and will positively influence your evaluation. Thank you for your consideration.

---

### Author Response · Authors · 2023-11-16
**Global Responses**

We extend our sincere gratitude to all the reviewers for their time and effort in reviewing our paper. Here, we respond to several common comments provided by the reviewers.

> Global Q1: 20 test rounds might be insufficient.

We sincerely appreciate the concerns of Reviewers WArX and 48Jp, it's important to note:

1. **The choice of 20 Avalon game rounds as a test size is substantial.** Each game encompasses numerous dialogues, making this a considerable volume of data. **A study recently accepted by EMNLP [1] introduced a dataset containing 20 Avalon games, which is equivalent to the number of test rounds per method in our study.** Hence, the test size per method in our research is comparable to a complete dataset in terms of volume. Additionally, **our experimental costs amounting to around $20,000 further emphasize the scale of the 20 Avalon games.**
2. **The 20 Avalon game rounds are adequate for demonstrating the efficacy of the ReCon framework and its components.** The high success rate depicted in Figure 4 of our paper, coupled with **the thorough statistical tests on our success rate improvement as elaborated in Global Q2 below**, further substantiates this claim.



[1] Stepputtis, Simon, et al. 'Long-Horizon Dialogue Understanding for Role Identification in the Game of Avalon with Large Language Models'. *Findings of the Association for Computational Linguistics: EMNLP*, 2023.



---



> Global Q2: It may be better to include statistical tests (or confidence intervals) to show the effectiveness of ReCon.

We are grateful for the input from Reviewers 48Jp and p3aL. **Following their suggestions, we applied Barnard's test [1] to the results in Figure 4 to determine if there is a statistically significant performance difference between our methods (ReCon and its ablated versions) and CoT.**

**The justification for selecting Barnard's test** is its suitability for analyzing 2x2 contingency tables, as described in [Wikipedia](https://en.wikipedia.org/wiki/Barnard's_test). This is relevant for our comparison of methods with CoT, where the numbers of successes and failures in each method form such contingency tables, making Barnard's test an appropriate choice.

The outcomes, specifically the p-values, are presented in the following table. **P-values less than 0.05, indicating statistical significance, are highlighted with an asterisk "*".**

| p-value                                      | ChatGPT                    | Claude     | Evil       |
| -------------------------------------------- | -------------------------- | ---------- | ---------- |
| ReCon w/o First-Order Perspective Transition | $0.0321^*$                 | $0.6497$   | $0.0114^*$ |
| ReCon w/o Second-Order Pespective Transition | $0.0427^*$                 | $0.3427$   | $0.0074^*$ |
| ReCon w/o Refinement Contemplation           | $0.0183^*$                 | $0.0830$   | $0.5484$   |
| ReCon w/o Formulation Contemplation          | $0.0005^*$                 | $0.5324$   | $0.0074^*$ |
| ReCon (Ours)                                 | ${3.4782\times 10^{-5}}^*$ | $0.0484^*$ | $0.0016^*$ |

The table above reveals that, except for their performance on Claude, nearly all our methods demonstrate statistically significant superiority over CoT, underscoring the effectiveness of our approaches in ReCon.

Moreover, **to further confirm the robustness of ReCon's designs on Claude, we increased the number of test games from approximately 20 to about 40.** This expansion aims to assess whether ReCon and its variations continue to outperform CoT with a larger set of test games. The results, displayed in the subsequent table, indicate that while success rates may vary with the number of games, **the findings observed with around 20 games remain consistent when increased to approximately 40 games. This consistency affirms the reliable effectiveness of ReCon's designs in the context of Claude.**

| Claude's success rate (v.s. baseline CoT)      | ~20 rounds | ~40 rounds |
| -------------------------------------------- | ---------- | ---------- |
| Baseline                                     | $47.4$%   | $45.0$%   |
| ReCon w/o First-Order Perspective Transition | $55.0$%   | $52.6$%   |
| ReCon w/o Second-Order Pespective Transition | $63.2$%   | $60.0$%   |
| ReCon w/o Refinement Contemplation           | $75.0$%   | $63.9$%  |
| ReCon w/o Formulation Contemplation          | $57.9$%  | $57.9$%   |
| ReCon (Ours)                                 | $78.9$%   | $73.7$%   |

[1] Fisher, Ronald Aylmer. "A new test for 2×2 tables." Nature 156.3961 (1945).

---

> ### Author Response · Authors · 2023-11-22
> **Assessment of GPT-4's Automated Evaluation by Human Evaluators**
>
> > Global Q3: How reliable is the GPT-4's automatic evaluation in the multi-dimensional evaluation part?
>
> **We assess GPT-4's automatic evaluation depicted in Figure 5 and Figure 6 through human annotations.** To do this, we select a random sample of 216 dialogues from those shown in Figure 5 and Figure 6. Each dialogue is classified according to the level of human consensus it receives, with the categories being "total agreement", "majority agreement", "majority disagreement", and "total disagreement". Our annotation team comprises 12 individuals, each responsible for 18 annotations, including 8 men and 4 women, all of whom have familiarity with the Avalon game. The outcomes of these annotations are as follows:
>
> |       | Full agreement          | Majority agreement | Majority disagreement      | Full disagreement | Total |
> | ----- | ----------------------- | ------------------ | -------------------------- | ----------------- | ----- |
> | Count | 56                      | 102                | 48                         | 10                | 216   |
> | Ratio | 25.93%                  | 47.22%             | 22.22%                     | 4.63%             | 100%  |
> |       | agreement ratio: 73.15% |                    | disagreement ratio: 26.85% |                   |       |
>
> Based on the data presented in the above table, it is evident that **the ratio of agreement significantly surpasses that of disagreement. This indicates that in our multi-dimensional evaluation, GPT-4's annotations are predominantly considered reliable.**
>
> Furthermore, to assess whether there was a significant difference in the frequency of "agreement" and "disagreement" ratings above, we conduct **the Chi-square test**. Our null hypothesis stated no difference in these frequencies. We observed 158 agreements and 58 disagreements out of 216 total responses. Under the null hypothesis, we expected equal frequencies for both categories (108 each). We calculated the Chi-square value (23.49) by comparing observed frequencies with expected frequencies. **The resulting p-value was about** $1.26\times 10^{-6}$**, far below the 0.05 threshold for statistical significance. This very low p-value led us to reject the null hypothesis, indicating a statistically significant difference between the counts of "agreement" and "disagreement" ratings.**